# Levels and determinants of health literacy and patient activation among multi-morbid COPD people in rural Nepal: Findings from a cross-sectional study

Uday Narayan Yadav[1,2,3]*, Jane Lloyd[1], Hassan Hosseinzadeh[4], Kedar Prasad Baral[5], Narendra Bhatta[6], Mark Fort Harris[1]

**1** Centre for Primary Health Care and Equity, UNSW, Sydney, Australia, **2** Forum for Health Research and Development, Dharan, Nepal, **3** School of Public Health and Community Medicine, UNSW, Sydney, Australia, **4** School of Health & Society, The University of Wollongong, Sydney, Australia, **5** School of Public Health, Patan Academy of Health Sciences, Kathmandu, Nepal, **6** Department of Pulmonary, Critical Care and Sleep Medicine, B.P. Koirala Institute of Health Sciences (BPKIHS), Dharan, Nepal

* u.yadav@unsw.edu.au, unyadav1@gmail.com

## Abstract

### Background

Health literacy (HL) and patient activation (PA) are necessary foundations to engage patients in self-management intervention. Each concept plays a unique role in improving access to the effective self-management of chronic disease. In this cross-sectional study, we examined the levels and determinants of HL and PA among the multi-morbid COPD patients in Nepal.

### Methods

We conducted interviews with a simple random sample of 238 multi-morbid COPD people from July 2018 to January 2019. The questionnaire included sociodemographic profiles, five domains of the Health Literacy Questionnaire (HLQ), 13-item Patient Activation Measure (PAM) and patient's illness perception by Brief Illness Perception Questionnaire (BIPQ). Multivariable logistic regression was used to examine the associations.

### Results

Most people with COPD had low health levels across each of the five domains of the HLQ. The proportion of people with low literacy level across each of the domains was: (i) feeling understood and supported by healthcare providers (79.0%), (ii) having sufficient information to manage my own health (76.5%), (iii) social support for health (77.3%), (iv) ability to find the good health information (75.2%), and (v) understand the health information well enough to know what to do (74.8%), respectively. The majority of patients also reported low levels of patient activation (level 1: 81.5%; level 2: 11.8%), with only 6.7% (level 3: 5%; level 4: 1.7%) reported higher patient activation level. We found significant associations between poor HL levels in the HLQ domains and having no education, being female or from *Indigenous* and

**Data Availability Statement:** All relevant data are within the manuscript and its Supporting Information files.

**Funding:** This fieldwork was funded by the Medibank International Fieldwork Grant and funding body had no role in the study.

**Competing interests:** The authors have declared that no competing interests exist.

**Abbreviations:** COPD, Chronic Obstructive Pulmonary Disease; HL, Health Literacy; PA, Patient Activation; SMPs, Self-management practices; PAM, Patient Activation Measure; PHC, Primary Health Care; HP, Health Post; HLQ, Health Literacy Questionnaire; HPS, feeling understood and supported by healthcare providers; HIS, having sufficient information to manage my own health; SS, social support for health; AE, ability to find the good health information; UHI, understand the health information well enough to know what to do; HREC, Human Research Ethics Committee (HREC); UNSW, University of New South Wales.

*Dalits* communities, and having a monthly family income of less than USD176. Having no education and poor illness perception were significantly associated with poor activation level on PAM scale.

## Conclusion

A high proportion of multi-morbid COPD peoples had low levels of HL and were less activated than what would be required to self-manage COPD. These were in turn associated with socioeconomic factors and poor illness perception. The findings from this study are being used to design a COPD self—management program tailored to the low health literate population.

## Background

COPD (Chronic Obstructive Pulmonary Disease) is a life-threatening lung disease characterised by persistent airflow limitation that interferes with normal breathing and is fully non-reversible [1]. Multi-morbidity among COPD patients contributes to the overall severity and increases the economic burden of treatment and health service costs [2, 3]. The World Health Organisation (WHO) reports that COPD caused 3.17 million (5%) deaths in 2015 world-wide and is anticipated to be in third position among all leading causes of death by 2020 [4]. The prevalence of COPD is increasing in Nepal. Subnational studies from Nepal reported the prevalence's of COPD ranging from 23% to 43% [5, 6].

Research has shown that improved health literacy has been associated with improvement in life style behaviours for chronic disease and decreased rates of hospitalisation [7, 8]. Quality of care for all chronic illness relies on informed, activated consumers and health care policy focusing on strategies related to informing and engaging patients [9].

The WHO defined health literacy as 'the personal characteristics and social resources needed for individuals and communities to access, understand, appraise and use information and services to make decisions about health' [10]. This definition broadens the concept of Health Literacy as 'skill set', by including consumers' motivation or 'mindset', which is crucial for engaging in healthy lifestyle behaviours [11]. The term 'patient and consumer activation/ motivation' is specifically defined as those who have the motivation, knowledge, skills and confidence to make effective decisions to manage their overall health [12], rather than focusing on a specific behaviour (e.g. Quitting smoking, getting involved in the physical activity) known as self-efficacy. Hibbard and colleagues developed a comprehensive scale to measure activation of patients for managing their health known as the Patient Activation Measure (PAM) [13]. Research has identified age, educational attainment, socioeconomic status, cultural beliefs and practices and communication (including language barriers) between professionals and patients as the determinants for low health literacy [14, 15]. Other determinants include symptom burden, illness perception, and presence of comorbidities, age, body mass index, physical health status, depression, social support, financial distress and lack of understanding their role in care process associated with lower patient activation [16, 17].

A study from a tertiary level hospital from eastern Nepal reported low levels of health literacy and knowledge related to chronic disease [18]. In this study, authors identified older age and being female, having low or no education, unemployment or retired status, poverty and history of smoking or consuming alcohol as being associated with inadequate health literacy.

Health literacy was found to be strong predictor of knowledge regarding hypertension, diabetes mellitus and COPD.

In recent years, the Government of Nepal has prioritised chronic conditions and is committed to deliver the services in line with National Multisectoral Action Plan(2014–2020) for the Prevention and Control of Non-Communicable Diseases [19]. One important objective of this plan was to prevent and control of NCDs by addressing the underlying social determinants of the health, through people-centred primary health care. In this light, emerging evidence [20] advocates that building health literacy could play an essential role in prevention, adherence to treatment, self-management, modification of unhealthy behaviours and the utilisation of the available health care services. HL and PA are important in supporting self-management practices for COPD patients where understanding of HL and PA levels is very important. There is no specific study that accessed the level of HL and PAM and their determinants focusing COPD patients from the community setting of Nepal. Our study aimed at examining the levels and determinants of HL and PA among the multi-morbid COPD patients in Nepal.

## Materials and methods

### Study design and participants

This was a community-based prospective cross-sectional study among multi-morbid COPD adults living in two rural municipalities of *Sunsari* district, Nepal. The study was carried out in between July 2018-January 2019. Simple random sampling was used to select the study participants from the study area. The sample size of 250 was calculated based on following assumptions: prevalence (knowledge on self-management) = 18% [21], sampling error = 5.0%, CI = 95.0%, and non-response rate = 10.0%. In the beginning, two Rural Municipalities (RMs) were randomly selected from the list of six RMs in *Sunsari* District. The list of the 576 individuals (296 from rural municipality A and 280 from rural municipality B considering the probability proportion to size) was prepared before conducting the field work, i.e., collecting the patient records from primary health care center (PHC) and health posts (HPs) of the study area. The data on people with COPD seeking service at private hospitals for COPD along with those seeking services private setting hospitals were collected with the help of Female Community Health Volunteers (FCHV), and the research assistant verified the data for such patients by looking through their medical records. Per protocol, 250 people with COPD were selected randomly from the list of eligible subjects. 238 people with COPD met the eligibility criteria for this study and were interviewed by the two trained interviewers in the community setting.

People with COPD were eligible to participate if they were adults aged between 18–70 years; had been diagnosed with COPD with at least one co-morbidity such as cardiovascular disease, diabetes, asthma, arthritis, depression, musculoskeletal disorders or gastritis in their medical records; and had been able to understand the information sheet or consent form. The informed consent form was read by the research assistants for those who were unable to read. All of the participants were informed that they were free to withdraw or opt out at any point during the interview. Prior to the interview, written informed consent was obtained from all literate participants, and thumb impressions were obtained from illiterate participants. Study participants with a hearing disability, severe cognitive disorder, kidney disease, a history of stroke or diagnosed heart attack and with terminal illness such as cancer were excluded from the study.

### Procedure

Interviewer-administered data collection was conducted over three months where participants provided information on socio-demographic items, a health literacy measure and, a measure

of patient activation. Two-day' hands-on training was provided to the RAs by one of the researchers of this study group which included field activities where they explained the questions and checked the understanding. In the pilot, we checked responses for inconsistency and did not find any evidence of inconsistency. The data collection was performed by trained RAs who were fluent in both Maithili (a local language spoken by chunks in a rural setting) and a Nepali language. The first author monitored the field in the community settings.

## Measurement

**Health literacy.** It was measured by using five of the nine domains of the Health Literacy Questionnaire (HLQ) [22]. The five domains were chosen based on relevance in the local context of Nepal. The HLQ domains were discussed between the research team and some of the experts from Nepal and the agreement was reached on using five domains. Of the excluded domain, three domains of HLQ were not relevant to Nepalese context and one domain was redundant because the PAM was also included. HLQ domains included in our study are: (i) feeling understood and supported by healthcare providers (HPS): four items, (ii) having sufficient information to manage my own health (HIS): four items, (iii) social support for health (SS: five items (iv) ability to find the good health information (AE): five items, and (v) understand the health information well enough to know what to do (UHI): five items. The logic behind the use of HLQ is that it measures the multidimensional health literacy profile, and the scales we used were culturally relevant to our setting. For the domains 1 to 3: participants responded as strongly disagree = 1, disagree = 2, agree = 3, strongly agree = 4 and for the domains 4 to 5: participant responded as cannot do = 1, very difficult = 2, quite difficult = 3, quite easy = 4, very easy = 5.The overall domain score was calculated by adding the item scores and then dividing by the number of items in that specific domain [23]. The cut-offs to define "high health literacy level" was determined considering upper quartile and the lower two quartiles as "low health literacy level". This applied to categorize all the five HLQ domains as there was no any standard cut-off.

**Patient activation.** The shortened Nepali version of the PAM (13-item) was used to access the self-reported knowledge, skill and confidence required for self-management of conditions [24]. This is a very reliable measure that captures aspects of motivation and engagement with self-management behaviors and health conditions. The 13 items have four possible responses option ranging from (i) strongly disagree to (iv) strongly agree, and an additional "not applicable" option and total scores of PAM ranges from 0–100. The raw score was calculated using a password protected scoresheet provided by Insignia Health and then excel data was imported in SPSS [25]. The raw scores for PAM were converted into four activation levels: (i) Level 1 (0.0–47.0): disengaged and overwhelmed, (ii) Level 2(47.1–55.1): becoming aware, but still struggling, (iii) Level 3(55.2–72.4): taking action and, (iv) Level 4(72.5–100.0): maintaining behaviors and pushing = more [25, 26]. We further categorized level 1 was as "poor activation stage" and Level 2 to 4 as "activation" stage.

**Independent variable measurement.** Independent variable included age; gender; marital status; religion; ethnicity; educational attainment; occupation status; income and, use of tobacco products. Similarly, the illness perception was measured using a nine-item scale brief illness perception questionnaire (BIPQ) [27] designed to assess the cognitive and emotional representations of illness rapidly. The higher the mean score of BIPQ, the greater negative reactions towards disease.

The English version of the socio-demographic profiles; HLQ and BIPQ were first translated to Nepali and then translated back to English by two external persons who were fluent in both Nepali and English language to check for consistency. The Nepali version of HLQ, BIPQ and

PAM were used to collect the information from the participants and these tools have been used previously [28–31] in Nepal. The Cronbach's alpha for each scale was found to be above 0.8.

### Ethics

The study received ethical approval from Human Research Ethics Committee (HREC) of University of New South Wales (UNSW), Sydney, Australia (HC180502), and the Nepal Health Research Council (Reg no 495). The study participants did not receive any benefits during their involvement in the research and written informed consent was obtained from all respondents prior to the interview.

### Statistical analyses

The statistical analyses were performed using the IBM Statistical Package for Social Sciences (SPSS 23.00) [32]. Participant characteristics were summarized as frequency and percentage. Normality of the data was assessed using Shapiro-Wilk test. Inter-quartile range was calculated for all included HLQ domains separately, where the upper quartile cut-offs was used to define "high health literacy level" and the lower two quartiles as "low health literacy level". The raw scores for PAM were categorized into four activation levels (Level 1, 2, 3 and 4) based on standard cut-off of PAM where level 1 was labelled as "poor activation stage" and Level 2 to 4 as "activation" stage. We used Chi-Square test for univariate analysis was conducted for all included five domains of HLQ and the PAM where all the independent variables that had $p < 0.2$ were considered in multivariable regression analysis. We assessed multicollinearity of covariates using Variance Inflation Factors (VIFs). The VIFs for all covariates that were included in the logistic regression analysis were less than 2.0. Backward elimination stepwise multivariable logistic regression analysis was performed to identify the determinants of poor HL and PA levels and the p-value ≤ 0.05 were considered as significant. The Hosmer-Lemeshow test was used for goodness of fit in the logistic regression model.

## Results

Overall, data of 238 participants were included in the analysis and are presented in Table 1. Study participants varied by gender, age group, education, marital status, the religion they practice, ethnicity, occupation, family level income, use of tobacco products and the presence of co-morbidities. Majority of tobacco users were found using smoking form of tobacco products like cigarettes, Bidis (small hand-rolled cigarettes made of tobacco and wrapped in leaf), Chillums (a straight conical pipe with end-to-end channel used for smoking).The majority of the participants had a high illness burden (BIPQ mean 49.83, SD = 8.86).

### Levels of HL and PA

Based on HLQ multi-dimensional scale, the proportion of people with low literacy level across the scales was: (i) HPS (79.0%), (ii) HIS (76.5%), (iii) SS (77.3%) (iv) AE (75.2%), and (v) UHI (74.8%),respectively. The majority of these patients reported low levels of patient activation (level 1, 81.5%; level 2, 11.8%), only 6.7% (level 3, 5%; level 4, 1.7%) reported higher patient activation level [Table 1].

The mean scores for HLQ domains and patient activation are presented in Table 2. The mean (SD) for HLQ domains: (i) HPS (1.96±.76), (ii) HIS (1.56±.74), (iii) SS (2.73±.77), (iv) AE (2.02±1.10), and (v) UHI (1.78±.99). The mean PAM score was 34.18 (14.20).

**Table 1. Characteristics of study samples.**

| Characteristics | n | % |
|---|---|---|
| **Gender** | | |
| Male | 107 | 45 |
| Female | 131 | 55 |
| **Age category (in years)** | | |
| ≤40 | 18 | 7.6 |
| 41–55 | 44 | 18.5 |
| ≥56 | 176 | 73.9 |
| **Education status** | | |
| Uneducated(not able to read and write) | 166 | 69.7 |
| Educated(able to read and write) | 72 | 30.3 |
| **Marital status** | | |
| Married | 176 | 73.9 |
| Unmarried/divorced/widow/widower | 62 | 26.1 |
| **Ethnicity** | | |
| Brahmin/Chhetri (Higher caste) | 98 | 41.2 |
| Indigenous | 91 | 38.2 |
| Dalits(untouchable caste as per traditional Hindu caste system) | 49 | 20.6 |
| **Religion** | | |
| Hindu | 222 | 93.3 |
| Islam | 16 | 6.7 |
| **Occupation** | | |
| Farmer/Housewife | 108 | 45.4 |
| Professional jobs (Business/government jobs/private company) | 8 | 3.4 |
| Factory workers/labourers | 122 | 51.3 |
| **Family income level**(1USD = 113.63 NPR) | | |
| ≤20,000 | 206 | 86.6 |
| >20000 | 32 | 13.4 |
| **Use of tobacco product***  | | |
| No | 45 | 18.9 |
| Yes | 193 | 81.1 |
| **Presence of co-morbidity** | | |
| At least one | 60 | 25.2 |
| Two or more | 178 | 74.8 |
| **Health literacy Questionnaire** | | |
| *HPS* | | |
| Low | 188 | 79.0 |
| High | 50 | 21.0 |
| *HIS* | | |
| Low | 182 | 76.5 |
| High | 56 | 23.5 |
| *SS* | | |
| Low | 184 | 77.3 |
| High | 54 | 22.7 |
| *AE* | | |
| Low | 179 | 75.2 |
| High | 59 | 24.8 |
| *UHI* | | |
| Low | 178 | 74.8 |
| High | 60 | 25.2 |

*(Continued)*

**Table 1.** (Continued)

| Characteristics | n | % |
|---|---|---|
| **Patient activation measure (PAM)** | | |
| Level 1 | 194 | 81.5 |
| Level 2 | 28 | 11.8 |
| Level 3 | 12 | 5 |
| Level 4 | 4 | 1.7 |
| **Total brief illness perception scale(n = 238)** | ------------------------------------ | |

HPS: Feeling understood and supported by healthcare providers.

HIS: Having sufficient information to manage my own health.

SS: Social support for health.

AE: Ability to find the good health information.

UHI: understand the health information well enough to know what to do.

## Associations with low HL scores across the HLQ domains

Table 2 presents univariate results for the associates of low HL and poor activation levels among our study samples.

Multivariate logistic regression analyses (Table 3) found significant associations between low health literacy level and socio-demographic variables: HPS domain was associated with having no education (AOR = 3.01, 95% CI: 1.44–6.29); HIS domain was associated with being female(AOR = 2.31, 95% CI: 1.02–5.23) or having no education (AOR = 3.11, 95% CI: 1.48–6.55); SS domain was associated with being *Indigenous* (AOR = 2.27, 95% CI: 1.14–4.50) or *Dalit* (AOR = 4.84, 95% CI: 1.57–14.83); AE domain was associated with being female (AOR = 2.56, 95% CI: 1.12–5.83), having no education (AOR = 6.40, 95% CI: 2.98–13.76), or having a family income level $\geq$ USD 176(AOR = 3.06, 95% CI: 1.17–8.04); UHI domain was associated with being female(AOR = 3.12, 95% CI: 1.28–7.59), or having no education (AOR = 7.06, 95% CI: 3.33–14.96) respectively.

## Associations with low PAM score

Poor activation (levels 1) was associated with having no education (AOR = 2.42, 95% CI: 1.09–5.38), or poor patients illness perception (AOR = 1.01, 95% CI: 1.00–1.11).

## Discussion

Health literacy [33] and activation levels [34] are foundational for the prevention and management of long-term conditions. The presence of co-morbidity among the chronic disease patients poses a significant and long-term challenge for patients, health services provider and the countries health care system. There is a scarce data on HL and PA among the multi-morbid COPD patients from the community setting in Nepal. A few studies have assessed HL, but none have measured both HL and PAM among chronic disease patients. To the best of our knowledge we are the first to examine the levels, relationship and determinants of health literacy and patient activation in multi-morbid COPD people in Nepal.

### PA and HL levels

In the present study three quarters of participants had low health literacy levels across the five domains of HLQ. Patients with lower HL had difficulty in feeling understood and supported

**Table 2. Univariate analysis to identify the determinants of health literacy questionnaire and PAM.**

| | HPS | | HIS | | SS | | AE | | UHI | | PAM | |
|---|---|---|---|---|---|---|---|---|---|---|---|---|
| | n(%) | | n(%) | | n(%) | | n(%) | | n(%) | | n(%) | |
| Variables | Low | High | Low | High | Low | High | Low | High | Low | High | No activation | Activation |
| **Gender** | 0.41 | | *0.07** | | 0.87 | | *<0.001*** | | *<0.001** | | *0.18**** | |
| Male | 82 | 25 | 76 | 31 | 82 | 25 | 68 | 39 | 68 | 39 | 83 | 24 |
| | (76.6) | (23.4) | (71.0) | (29.0) | (76.6) | (23.4) | (63.6) | (36.4) | (63.6) | (36.4) | (77.6) | (22.4) |
| Female | 25 | 106 | 106 | 25 | 102 | 29 | 111 | 20 | 110 | 21 | 111 | 20 |
| | (19.1) | (80.9) | (80.9) | (19.9) | (77.9) | (22.1) | (84.7) | (15.3) | (84.0) | (16.0) | (84.7) | (15.3) |
| **Age category(in years)** | 0.55 | | *0.03** | | 0.71 | | *0.04** | | *0.05** | | *0.18**** | |
| ≤40 | 13 | 5 | 11 | 7 | 14 | 4 | 13 | 5 | 10 | 8 | 13 | 5 |
| | (72.2) | (27.8) | (61.1) | (38.9) | (77.8) | (22.2) | (72.2) | (27.8) | (55.6) | (44.4) | (72.2) | (27.8) |
| 41–55 | 33 | 11 | 29 | 15 | 36 | 8 | 27 | 17 | 30 | 14 | 33 | 11 |
| | (75.0) | (25.0) | (65.9) | (34.1) | (81.8) | (18.8) | (61.4) | (38.6) | (68.2) | (31.8) | (75.0) | (25.0) |
| ≥56 | 142 | 34 | 142 | 34 | 134 | 42 | 139 | 37 | 138 | 38 | 148 | 28 |
| | (80.7) | (19.3) | (80.7) | (19.3) | (76.1) | (23.9) | (79.0) | (21.0) | (78.4) | (21.6) | (84.1) | (15.9) |
| **Education status** | *<0.001*** | | *<0.001*** | | *0.02** | | *<0.001*** | | *<0.001*** | | *<0.001*** | |
| Uneducated | 143 | 23.0 | 141 | 25 | 135 | 31 | 147 | 19 | 148 | 18 | 145 | 21 |
| | (86.1) | (13.9) | (84.9) | (15.1) | (81.3) | (18.7) | (88.6) | (11.4) | (89.2) | (10.8) | (87.3) | (12.7) |
| Educated | 45 | 27 | 41 | 31 | 49 | 23 | 32 | 40 | 30 | 42 | 49 | 23 |
| | (62.5) | (37.5) | (56.9) | (43.1) | (68.1) | (31.9) | (44.4) | (55.6) | (41.7) | (58.2) | (68.1) | (31.9) |
| **Marital status** | *<0.001*** | | *0.02** | | 0.71 | | *<0.001*** | | *<0.001*** | | *<0.001*** | |
| Married | 131 | 45 | 128 | 48 | 135 | 41 | 124 | 52 | 122 | 54 | 139 | 37 |
| | (74.4) | (25.6) | (72.7) | (27.3) | (76.7) | (23.3) | (70.5) | (29.5) | (69.3) | (30.7) | (79.0) | (21.0) |
| Others | 57 | 5 | 54 | 8 | 49 | 13 | 55 | 7 | 56 | 6 | 55 | 7 |
| | (91.9) | (8.1) | (87.1) | (12.9) | (79.0) | (21.0) | (88.7) | (11.3) | (90.3) | (9.7) | (88.7) | (11.3) |
| **Ethnicity** | 0.56 | | 0.96 | | *<0.001*** | | 0.28 | | 1.00 | | *0.16**** | |
| Higher Caste | 78 | 20 | 75 | 23 | 65 | 33 | 73 | 25 | 73 | 25 | 85 | 13 |
| | (79.6) | (20.4) | (76.5) | (23.5) | (66.3) | (33.7) | (74.5) | (25.5) | (74.5) | (25.5) | (86.7) | (13.3) |
| Indigenous | 69 | 22 | 69 | 22 | 74 | 17 | 65 | 26 | 68 | 23 | 69 | 22 |
| | (75.8) | (24.2) | (75.8) | (24.2) | (81.3) | (18.7) | (71.4) | (28.6) | (74.7) | (25.3) | (75.8) | (24.2) |
| Dalits | 41 | 8 | 38 | 11 | 45 | 4 | 41 | 8 | 37 | 12 | 40 | 9 |
| | (83.7) | (16.3) | (77.6) | (22.4) | (91.8) | (8.2) | (83.7) | (16.3) | (75.5) | (24.5) | (81.6) | (18.4) |
| **Occupation** | *<0.001*** | | *<0.001*** | | 0.25 | | *<0.001*** | | *<0.001*** | | *<0.001*** | |
| Professionals(Government jobs, Private, Businessman) | 4 | 4 | 4 | 4 | 5 | 3 | 3 | 5 | 3 | 5 | 5 | 3 |
| | (50.0) | (50.0) | (50.0) | (50.0) | (62.5) | (37.5) | (37.5) | (62.5) | (37.5) | (62.5) | (62.5) | (37.5) |
| Farmer/housewife | 74 | 34 | 70 | 38 | 88 | 20 | 75 | 33 | 73 | 35 | 76 | 32 |
| | (68.5) | (38.5) | (64.8) | (35.2) | (81.5) | (18.5) | (69.4) | (30.6) | (67.6) | (32.4) | (70.4) | (29.6) |
| Factory workers/labourers | 110 | 12 | 108 | 14 | 91 | 31 | 101 | 21 | 102 | 20 | 113 | 9 |
| | (90.2) | (9.8) | (88.5) | (11.5) | (74.6) | (25.4) | (82.8) | (17.2) | (83.6) | (16.4) | (92.6) | (7.4) |
| **Family level income** | *0.04** | | 0.51 | | *0.04** | | *<0.001*** | | *0.04** | | 0.59 | |
| <20,000 | 167 | 39 | 159 | 47 | 164 | 42 | 162 | 44 | 159 | 47 | 169 | 37 |
| | (81.1) | (18.9) | (77.2) | (22.8) | (79.6) | (20.4) | (78.6) | (21.4) | (77.2) | (22.8) | (82) | (18) |
| >20,000 | 21 | 11 | 23 | 9 | 20 | 12 | 17 | 15 | 19 | 13 | 25 | 7 |
| | (65.6) | (34.4) | (71.9) | (28.1) | (62.5) | (37.5) | (53.1) | (46.9) | (59.4) | (40.6) | (78.1) | (21.9) |
| **Use of tobacco products** | *0.07** | | *0.01** | | 0.66 | | 0.57 | | *0.08** | | *0.13**** | |
| No | 31 | 14 | 28 | 17 | 35 | 10 | 32 | 13 | 29 | 16 | 161 | 32 |
| | (68.9) | (31.1) | (62.2) | (37.8) | (77.8) | (22.2) | (71.1) | (28.9) | (64.4) | (35.6) | (83.4) | (16.6) |

(*Continued*)

**Table 2.** (Continued)

| Variables | HPS n(%) | | HIS n(%) | | SS n(%) | | AE n(%) | | UHI n(%) | | PAM n(%) | |
|---|---|---|---|---|---|---|---|---|---|---|---|---|
| | Low | High | Low | High | Low | High | Low | High | Low | High | No activation | Activation |
| Yes | 157 | 36 | 154 | 39 | 149 | 44 | 147 | 46 | 149 | 44 | 33 | 12 |
| | (81.3) | (18.7) | (79.8) | (20.2) | (77.2) | (22.8) | (76.2) | (23.8) | (77.2) | (22.8) | (17.3) | (26.7) |
| **Presence of co-morbidity** | 0.36 | | *0.14*\*\*\* | | 0.21 | | 1 | | 0.74 | | 0.45 | |
| At least one | 138 | 40 | 132 | 46 | 134 | 44 | 134 | 44 | 132 | 46 | 143 | 35 |
| | (77.5) | (22.5) | (74.2) | (25.8) | (75.3) | (24.7) | (75.3) | (24.7) | (74.2) | (25.8) | (80.3) | (19.7) |
| Two or more | 50 | 10 | 50 | 10 | 50 | 10 | 45 | 15 | 46 | 14 | 51 | 5 |
| | (83.3) | (16.3) | (83.3) | (16.7) | (83.3) | (16.7) | (75.0) | (25.0) | (76.7) | (23.3) | (85.0) | (15.0) |
| Total illness perception scale (Mean± SD) | 49.83±8.86 | | | | | | | | | | | |

p-value\*\*\*<0.2, p-value\* <0.05, p-value\*\* <0.001.

HPS: Feeling understood and supported by healthcare providers.

HIS: Having sufficient information to manage my own health.

SS: Social support for health.

AE: Ability to find the good health information.

UHI: understand the health information well enough to know what to do.

by healthcare providers, had insufficient information to manage their own health, were unable to find good health information or understand health information well enough to know what to do. If their level of health literacy is low, people may not be able to navigate the health system. Moreover, they lack the knowledge, may be misinformed about the cause of disease or may not understand the importance of relationship between lifestyle behaviours and health outcomes.

A recently published study from Nepal conducted among chronic disease patients recruited from tertiary level hospital and a community hospital using Europe-Asia Health Literacy Survey Questionnaire showed 19% had marginal HL and 54% had inadequate HL [18]. Overall, a higher proportion of patients had low literacy in our study population when compared with studies from India [35] and Bangladesh [36]. Similarly, the estimate is higher than that reported among COPD patients in the United Kingdom (15%) [37], Spain(59%) [38], Europe (47%) [39] or the United States of America (29%) [40]. The higher estimate of low HL could be explained by a larger proportion of the participants had no formal education and also had limited right information about the availability of health services for respiratory conditions at a primary health care or other health care setting in Nepal.

Similarly, a high proportion of multi-morbid COPD patients demonstrated a low level (level 1:81.5%) of activation. We could not find any study from Nepal on PAM levels to compare with our study findings. A hospital-based study from India has shown that nearly 50% (level 1:21.88, level 2: 25.57) of the sample were at a low level of activation [41]. In contrast to our findings, studies from the Australia [42], Netherlands [43], and USA [17] reported lower proportions of less activated patients. In this study, the mean scores on the PAM-13 were much lower than aforementioned studies from India, Australia, Netherlands and USA. In our study, a lower level of activation might be explained by participants having one or more multi-morbid conditions. The low activation level is plausible; given that in the rural part of Nepal where illiteracy is high, people with chronic disease might not have received adequate support from family members/care givers and health care professionals to manage their conditions. More importantly, activation might have differed across gender, ethnicity, age, education and income level and few of these have been reported to be correlated with activation levels in

**Table 3. Stepwise multivariable logistic regression analysis for low health literacy and a poor activation stage on PAM.**

| Variables | AOR(95% CI) | | | | | |
|---|---|---|---|---|---|---|
| | **HPS** | **HIS** | **SS** | **AE** | **UHI** | **PAM** |
| **Gender** | | | | | | |
| Male | NI | 1 | NI | 1 | 1 | 1 |
| Female | | **2.31(1.02–5.23)** | | **2.56(1.12–5.83)** | **3.12(1.28–7.59)** | 2.04(.86–4.80) |
| **Age category (in years)** | | | | | | |
| ≤40 | NI | 1 | NI | 1 | 1 | 1 |
| 41–55 | | .73(.18–2.86) | | .51(.12–2.17) | 1.50(.35–6.33) | .89(.21–3.73) |
| ≥56 | | .68(.27–1.67) | | 1(.25–3.98) | 1.26(.30–5.30) | .89(.20–3.90) |
| **Education status** | | | | | | |
| Educated | **1** | **1** | 1 | **1** | **1** | **1** |
| Uneducated | **3.01(1.44–6.29)** | **3.11(1.48–6.55)** | 1.79(.91–3.12) | **6.40(2.98–13.76)** | **7.06(3.33–14.96)** | **2.42(1.09–5.38)** |
| **Marital status** | | | | | | |
| Married | 1 | 1 | NI | 1 | 1 | 1 |
| Others | 2.63(.94–7.35) | 1.45(.58–3.58) | | .49(.18–1.33) | .38(.13–1.10) | 1.05(.39–2.85) |
| **Ethnicity** | | | | | | |
| Higher Caste | NI | NI | 1 | | 1 | 1 |
| Indigenous | | | **2.27(1.14–4.50)** | .85(.38–1.90) | .55(.18–1.61) | .57(.25–1.32) |
| Dalits | | | **4.84(1.57–14.83)** | 1.06(.37–3.07) | .52(.12–2.20) | .37(.13–1.1) |
| **Occupation** | | | | | | |
| Professionals(Government jobs, Private, Businessman) | 1 | 1 | NI | 1 | 1 | 1 |
| Farmer/housewife | 1.34(.26–6.79) | .58(.12–2.90) | | 1.21(.21–7.01) | .95(.17–5.29) | .80(.16–3.92) |
| Factory workers/labourers | 4.61(.82–25.72) | 2.16(.41–11.32) | | 2.16(.36–12.93) | 2.36(.41–13.58) | 4.6(.85–25.42) |
| **Family level income** | | | | | | |
| <20,000 | 1.84(.74–4.52) | NI | 1.87(.81–4.31) | **3.06(1.17–8.04)** | 2.06(.78–5.44) | NI |
| >20,000 | | | 1 | 1 | 1 | |
| **Use of tobacco products** | | | | | | |
| No | 1 | 1 | NI | NI | 1 | 1 |
| Yes | 1.31(.58–2.97) | 2.25(.86–5.86) | | | 2.04(.70–5.96) | 1.78(.62–5.09) |
| **Presence of co-morbidity** | | | | | | |
| At least one | NI | 2.25(.93–5.46) | NI | NI | NI | NI |
| Two or more | | 1 | | | | |
| **Total illness perception scale** | .99(.96–1.03) | .98(.95–1.02) | .99(.96–1.03) | .99(.95–1.03) | 1.01 (.97–1.05) | **1.01(1.00–1.11)** |

Model is adjusted for the all variables that are considered in each model.

NI: not included in the particular model.

HPS: Feeling understood and supported by healthcare providers.

HIS: Having sufficient information to manage my own health.

SS: Social support for health.

AE: Ability to find the good health information.

UHI: understand the health information well enough to know what to do.

previous studies [17, 44]. Moreover, social-desirable bias might have leaded to overstate responses.

## Associates of HL and PAM

Importantly, having no education was significantly associated with low HL levels across four domains of the HLQ. Patients with poor levels of education find difficulty in understanding

the information provided by health care providers, don't have enough information for management of their conditions and, have poorer ability to find and understand health information. In the Nepalese context, many uneducated people are indiscriminate in their search for information about the conditions and medications, delay seeking the available health services and find difficulty communicating with health professionals about their health conditions. They often they seek information from other educated people, teachers and mid-level health professionals. Having no education deprives them of getting better jobs, having a good income and thus limits their timely access to health service which in turn may lead to poor health actions required for managing conditions. Previous studies support our findings on educational attainment [40, 45, 46] as an important determinant for lower health literacy. Thus, there is need to close the gap between what healthcare professionals knows and provide, and what the people with no education understands and use the information for decision making. Adopting teach-back process could be a good way to address the HL at the care delivery level (primary health care centres, clinics or hospital) so that patients don't miss appointment, can have enough knowledge of disease and healthy lifestyle habits, and be able to understand the medications and the purpose of taking medications.

Consistent with other research, we found that being female [47, 48] was a determinant of lower HL level across three domains of HL (HIS, AE, UHI). In our study, females were less empowered with their role restricted to household activities. Women in rural area (particularly in rural Terai region) have less autonomy in decision making about their own health because of unequal access to education and health care [49]. This study was conducted in the plains region of Nepal, where the majority of populations are unprivileged (Madeshi, Indigenous and Dalits origin), where women do not have the ability to find sufficient health information. Thus, an approach that is gender based may be required to improve patient health literacy. Another study from Nepal supports our finding that being female was associated with poor health literacy [18, 50]. In contrast, evidence from China did not find a relationship between gender and HL while a study from the US, showed females had higher HL, possible due the difference in socio-demographic profiles of the studied populations [11, 51]. In line with our findings, published work has shown that not being married or divorced were associated with poor health literacy [40]. Interestingly, families with low incomes were more likely to have low HL in the AE domain of the HLQ. This could be because the patients from low-income level family might not seek health services and ignore the health problems because of the financial barrier to their access. Moreover, ability to find good information about the conditions and available services may not be a priority in comparison with other daily living needs such as employment, farming etc. Supporting our findings, other authors have reported family income to be a determinant for HL [40, 44, 51].

More than half of the participants who were from *Indigenous and Dalits* community were more likely to have poor health literacy on the SS domain. The one potential explanation could be poor socio-economic status of the family/caregivers in this community. For instance, a financially challenged caregiver might not be able to fulfil the daily needs and take a proper care of the health of people with chronic diseases such as COPD. Furthermore, the poor socio-cultural environment of the disadvantaged ethnic groups may result in caregivers being unwilling to listen to the health problems of the older family member with condition like COPD and to provide them with the support they need [52–54].

We found that having no education and poor illness perception was associated with poor levels of activation. This is supported by other evidence demonstrating that being uneducated was associated with lower PAM [44]. Association between illness perception and PAM scores was also supported by the findings of other studies [16, 17]. A high illness perception score reflects patients 'experience of fear, threat, breathlessness and poor emotional responses to the

disease. This might have a significant impact on the quality of life of people with COPD [55]. Thus, we suggest the need to address illness perception as an important predictor for activating the patients to engage in self-management of behaviours.

In sum up, our findings underscore the need for a program that is tailored to patient HL and activation levels, and which supports or develops decision-making, goal setting and self-management skills. Such a program must include activities to increase awareness of disease and conditions, improve communication skills with health professionals. It also needs to improve access to quality information on their conditions and medications, diet and exercise and creates an enabling environment. Moreover, health care providers need to be aware that HL and PA are different but correlated and must both taken into consideration while planning care. These results provide compelling evidence to integrate HL and PA initiatives in all health policies aimed at improving the behaviour change required for self-management of the conditions in this population.

Strength of this study is that it is one of the first studies of its kind to assess the HL and PA among the multi-morbid COPD patients in rural Nepal. A limitation is that the study was conducted in two rural municipalities of the Sunsari district, thus, the results cannot be generalised to other settings of Nepal. Additionally, we have relied on government data and medical records of patients and have not verified the cases with any objective measures. Further, this study was cross-sectional in nature and thus the causal relationship between the dependent and independent variables cannot be established.

## Conclusion

In conclusion, a high proportion of multi-morbid COPD patients had low levels of HL and were less activated that was required for self-management of COPD. Having no education, being female, a monthly family income less than USD176 and being from indigenous and Dalit communities, were associated with lower level of health literacy. Low education attainment and poor illness perception were associated with being less activated in their care. Addressing both levels and determinants of HL and patient activation is required to improve self-management practices among COPD patients.

## Supporting information

**S1 Data.**
(SAV)

## Acknowledgments

We would like to thank the local-level authorities of the government of Nepal for their support in providing us with the list of COPD patients. The authors like to thank, more respectfully to Mr Gajendra Yadav (Health Co-ordinator of Gadhi Rural Municipality), Mr Dinesh Chaudhary (Barju Rural Municipality) for providing the support in reaching the patients and, Mr Krishna Yadav and Nitesh Mandal for their tireless work in recruitment and data collection. The COPD patients who participated in the study are also gratefully acknowledged without which this study would not have been successful. UNY is receipt of International Postgraduate Scholarship and CPHCE Top-up Scholarship for pursuing PhD.

## Author Contributions

**Conceptualization:** Uday Narayan Yadav, Jane Lloyd, Hassan Hosseinzadeh, Mark Fort Harris.

**Data curation:** Uday Narayan Yadav, Jane Lloyd, Hassan Hosseinzadeh, Mark Fort Harris.

**Formal analysis:** Uday Narayan Yadav, Jane Lloyd, Hassan Hosseinzadeh, Mark Fort Harris.

**Funding acquisition:** Uday Narayan Yadav, Jane Lloyd, Hassan Hosseinzadeh, Mark Fort Harris.

**Investigation:** Uday Narayan Yadav, Jane Lloyd, Hassan Hosseinzadeh, Kedar Prasad Baral, Narendra Bhatta, Mark Fort Harris.

**Methodology:** Uday Narayan Yadav, Jane Lloyd, Hassan Hosseinzadeh, Narendra Bhatta, Mark Fort Harris.

**Project administration:** Uday Narayan Yadav, Jane Lloyd, Hassan Hosseinzadeh, Kedar Prasad Baral, Narendra Bhatta, Mark Fort Harris.

**Resources:** Uday Narayan Yadav, Jane Lloyd, Hassan Hosseinzadeh, Mark Fort Harris.

**Software:** Uday Narayan Yadav, Jane Lloyd, Hassan Hosseinzadeh, Mark Fort Harris.

**Supervision:** Uday Narayan Yadav, Jane Lloyd, Hassan Hosseinzadeh, Kedar Prasad Baral, Narendra Bhatta, Mark Fort Harris.

**Validation:** Uday Narayan Yadav, Jane Lloyd, Hassan Hosseinzadeh, Kedar Prasad Baral, Narendra Bhatta, Mark Fort Harris.

**Visualization:** Uday Narayan Yadav, Jane Lloyd, Hassan Hosseinzadeh, Narendra Bhatta, Mark Fort Harris.

**Writing – original draft:** Uday Narayan Yadav.

**Writing – review & editing:** Uday Narayan Yadav, Jane Lloyd, Hassan Hosseinzadeh, Kedar Prasad Baral, Narendra Bhatta, Mark Fort Harris.

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
