## [Decision Letter · Decision Letter 0]

27 Oct 2019

PONE-D-19-18916

Levels and determinants of health literacy and patient activation in multimorbid COPD patients in rural Nepal: Findings from a cross-sectional study

PLOS ONE

Dear Mr Yadav,

Thank you for submitting your manuscript to PLOS ONE. After careful consideration, we feel that it has merit but does not fully meet PLOS ONE’s publication criteria as it currently stands. Therefore, we invite you to submit a revised version of the manuscript that addresses the points raised during the review process.

The reviewers find the work of merit but have requested some additions and revisions, In addition to the items raised by the reviewers, please address the following points:

Please clarify scoring and categorization of HQL and PAM in more detail. We used the PAM questionnaire as an instrument to measure Customer Quality in our work as a third dimension to quality. You can find more detail in "Gholipour K, Tabrizi JS, Jafarabadi MA, Iezadi S, Mardi A. Effects of customer self-audit on the quality of maternity care in Tabriz: A cluster-randomized controlled trial. PloS one. 2018 Oct 11;13(10):e0203255." How you categorized PAM result in four level?

Provide detailed reference for SPSS as " IBM SPSS Statistics for Windows, version 23 (IBM Corp., Armonk, N.Y., USA)".

Please report regression prerequisite in method section and its handling methods.

How you test the normality of data to use T-test and ANOVA? What test did you used in case the data were not normal?

A P-value was duplicated in table 2.

Provide B: Unstandardized coefficients, S.E.: Standard error and Beta: Standardized regression coefficients for variable and model Goodness of Fit in table 3.

We would appreciate receiving your revised manuscript by Dec 11 2019 11:59PM. To enhance the reproducibility of your results, we recommend that if applicable you deposit your laboratory protocols in protocols.io, where a protocol can be assigned its own identifier (DOI) such that it can be cited independently in the future. For instructions see: http://journals.plos.org/plosone/s/submission-guidelines#loc-laboratory-protocols

We look forward to receiving your revised manuscript.

Kind regards,

Kamal Gholipour, PhD

Academic Editor

PLOS ONE

Journal Requirements:

Reviewers' comments:

Reviewer's Responses to Questions

**Comments to the Author**

1. Is the manuscript technically sound, and do the data support the conclusions?

Reviewer #1: Yes

Reviewer #2: Yes

2. Has the statistical analysis been performed appropriately and rigorously? 

Reviewer #1: No

Reviewer #2: I Don't Know

3. Have the authors made all data underlying the findings in their manuscript fully available?

Reviewer #1: Yes

Reviewer #2: Yes

4. Is the manuscript presented in an intelligible fashion and written in standard English?

Reviewer #1: Yes

Reviewer #2: Yes

5. Review Comments to the Author

Reviewer #1: In this cross-sectional study, the authors examined the levels and determinants of HL and PA among the multi-morbid COPD patients in Nepal. They found that a high proportions of multi-morbid COPD patients had low levels of HL and were less activated than what would be required to self-manage COPD. The results may be help to design a family and patient COPD self - management intervention tailored to a low health literate population. Here are some comments for next revision.

1. Multivariate logistic regression is better statistical analyses in this research than multiple linear regression. It's suggest to analysis with multivariate logistic regression and compare the results.

2. We know that the conditions used by the Independent t-test and One-way Analysis of Variance (ANOVA) are that the data should conform to the normal distribution and the homogeneity of the variance. Has the author conducted the test? If not, use the rank sum test please.

3. The Illness Perception scale indicator is not statistically analyzed in Tables 1 and 2. Please explain why is it included in Table 3.

4. Page 13, the authors describe the majority of the participants had a high of illness burden (BIPQ mean 49.83, SD= 8.86), but it is unclear as to the purpose of including this number.

5. Page 14, the authors do not provide information on whether the overall model was significant or accompanying descriptions of effect size, For example: Square value of R.

Reviewer #2: The authors present a paper addressing the levels and determinants of health literacy and patient activation in multimorbid COPD patients in rural Nepal. Health literacy is a very important subject to reduce the impact of chronic diseases in health systems. The results are relevant to the community chosen for this study. Although the subject is interesting the paper does not read well. There are misspellings throughout the text that should be corrected. In measurement section the text is not clear. What do the authors mean by items?

COPD should be in full in abstract.

6. PLOS authors have the option to publish the peer review history of their article (what does this mean?). If published, this will include your full peer review and any attached files.

Reviewer #1: No

Reviewer #2: Yes: Monica Botelho

---

## [Author Response · Author response to Decision Letter 0]

29 Nov 2019

Dear Editor, Date: 29/11/2019

I would take this opportunity to thank you and the reviewers for your valuable remarks. We authors have addressed all the remarks and have improved the quality of paper. In particular, we revised our analysis using logistic regression method.

Once again, many thanks for your valuable remarks.

Looking forward to hearing from you.

Warm Regards,

Uday

Editor remarks 

Please clarify scoring and categorization of HQL and PAM in more detail. We used the PAM questionnaire as an instrument to measure Customer Quality in our work as a third dimension to quality. You can find more detail in "Gholipour K, Tabrizi JS, Jafarabadi MA, Iezadi S, Mardi A. Effects of customer self-audit on the quality of maternity care in Tabriz: A cluster-randomized controlled trial. PloS one. 2018 Oct 11;13(10):e0203255." How you categorized PAM result in four level?

 Thanks much for sharing the wonderful piece and critical look. We have added the cut-off in the updated document.

Health Literacy

The cut-offs to define “high health literacy level” was determined considering upper quartile and the lower two quartiles as “low health literacy level”. This applied to categorize all the five HLQ domains as there was no any standard cut-off.[Line 186-188]

PAM

The raw score was calculated using password protected scoresheet provided by Insignia Health and then excel data was imported in SPSS.(21). The raw scores of PAM were converted into four activation levels: (i) Level 1(0.0-47.0): disengaged and overwhelmed, (ii) Level 2(47.1-55.1): becoming aware, but still struggling, (iii) Level 3(55.2-72.4): taking action and, (iv) Level 4(72.5-100.0): maintaining behaviors and pushing further.(21, 22) 

[Line 196-200]

Provide detailed reference for SPSS as " IBM SPSS Statistics for Windows, version 23 (IBM Corp., Armonk, N.Y., USA)".

 We have included in the revised version.

Please report regression prerequisite in method section and its handling methods.

 Based on reviewer and editor remarks, we checked the assumptions of linear regression and we revised our analysis and have used logistic regression.

The statistical analyses were performed using the Statistical Package for Social Sciences (SPSS 23.00)(25).Participant characteristics were summarised as frequency and percentage. Normality of the data was assessed using Shapiro-Wilk test. We used Chi-Square test for univariate analysis and all the independent variables that had p<0.2 were considered in multivariable regression analysis. Step-wise multivariable logistic regression analysis was used to identify the determinants of poor HL and PA levels and the p-value ≤ 0.05 were considered as significant.

How you test the normality of data to use T-test and ANOVA? What test did you used in case the data were not normal?

 The Shapiro-Wilks test was used to investigate the normality of the data and we found no violations of regression assumptions. A significance level of p<0.05 for all the independent variables with dependent variables (HLQ domains and PAM) in univariate analysis were considered in multivariable regression analysis. Multivariable linear regression analysis was used to assess associations of the independent variable with scores of HLQ scales and PAM separately. [Page 7]

A P-value was duplicated in table 2.

Provide B: Unstandardized coefficients, S.E.: Standard error and Beta: Standardized regression coefficients for variable and model Goodness of Fit in table 3.

 Analysis is changed in the revised version.

Reviewer 1 

Multivariate logistic regression is better statistical analyses in this research than multiple linear regression. It's suggest to analysis with multivariate logistic regression and compare the results. Dear Reviewer,

Thanks for your suggestion and we have changed our analysis. Much appreciated for this.

We know that the conditions used by the Independent t-test and One-way Analysis of Variance (ANOVA) are that the data should conform to the normal distribution and the homogeneity of the variance. Has the author conducted the test? If not, use the rank sum test please.- 

 Analysis is changed in the revised version.

We used Chi-Square test for univariate analysis and all the independent variables that had p<0.2 were considered in multivariable regression analysis. Step-wise multivariable logistic regression analysis was used to identify the determinants of poor HL and PA levels

The Illness Perception scale indicator is not statistically analysed in Tables 1 and 2. Please explain why it included in Table 3 is.

4. Page 13, the authors describe the majority of the participants had a high of illness burden (BIPQ mean 49.83, SD= 8.86), but it is unclear as to the purpose of including this number.

 We have included in results section as: The majority of the participants had a high of illness burden (BIPQ mean 49.83, SD= 8.86). 

Thank you for very much for your sharp look. 

In this revised version we have included it Table1,2 and Table 3.

Page 14, the authors do not provide information on whether the overall model was significant or accompanying descriptions of effect size, For example: Square value of R. Analysis is changed in the revised version.

Models are adjusted for all the included variables in logistic regression.

Reviewer 2 

The authors present a paper addressing the levels and determinants of health literacy and patient activation in multimorbid COPD patients in rural Nepal. Health literacy is a very important subject to reduce the impact of chronic diseases in health systems. The results are relevant to the community chosen for this study. Although the subject is interesting the paper does not read well. There are misspellings throughout the text that should be corrected. In measurement section the text is not clear. What do the authors mean by items? Thanks much for noting our misspellings. We have worked on the misspellings throughout the paper. Items mean the questions in each domain.

---

## [Decision Letter · Decision Letter 1]

30 Jan 2020

PONE-D-19-18916R1

Levels and determinants of health literacy and patient activation among multi-morbid COPD people in rural Nepal: Findings from a cross-sectional study

PLOS ONE

Dear Mr Yadav,

Thank you for submitting your manuscript to PLOS ONE. After careful consideration, we feel that it has merit but does not fully meet PLOS ONE’s publication criteria as it currently stands. Therefore, we invite you to submit a revised version of the manuscript that addresses the points raised during the review process.

The reviewers find the work of merit but have requested some additions and revisions.==============================

We would appreciate receiving your revised manuscript by Mar 15 2020 11:59PM. To enhance the reproducibility of your results, we recommend that if applicable you deposit your laboratory protocols in protocols.io, where a protocol can be assigned its own identifier (DOI) such that it can be cited independently in the future. For instructions see: http://journals.plos.org/plosone/s/submission-guidelines#loc-laboratory-protocols

We look forward to receiving your revised manuscript.

Kind regards,

Kamal Gholipour, PhD

Academic Editor

PLOS ONE

Reviewers' comments:

Reviewer's Responses to Questions

**Comments to the Author**

1. If the authors have adequately addressed your comments raised in a previous round of review and you feel that this manuscript is now acceptable for publication, you may indicate that here to bypass the “Comments to the Author” section, enter your conflict of interest statement in the “Confidential to Editor” section, and submit your "Accept" recommendation.

Reviewer #3: (No Response)

2. Is the manuscript technically sound, and do the data support the conclusions?

Reviewer #3: Partly

3. Has the statistical analysis been performed appropriately and rigorously? 

Reviewer #3: No

4. Have the authors made all data underlying the findings in their manuscript fully available?

Reviewer #3: Yes

5. Is the manuscript presented in an intelligible fashion and written in standard English?

Reviewer #3: Yes

6. Review Comments to the Author

Reviewer #3: The manuscript entitled 'Levels and determinants of health literacy and patient activation among multi-morbid COPD people in rural Nepal: Findings from a cross-sectional study' with the aim to examine the levels and determinants of HL and PA among the multi-morbid COPD patients in Nepal.

The manuscript requires further improvement.

Comments

Page 4 Line 143, the sample size calculation was based on knowledge of self-management. What was the prevalence figure contributed by HL and PA respectively?

Page 5, for the subjects selection into the study what was the minimum education attainment and how to ensure that the subjects understand the study and know what they have been asked?

Page 5 Line 145, the total number of RMs in Sunsari District to be stated.

Page 5 Line 158, what about the level of understanding on the subject matter of the study?

Page 5 Line 159-260, the sentence can be further improved.

Page 6 Line 210, brief IPQ to be written as BIPQ.

Questionnaires

There was no information if the Nepalese version questionnaires (HLQ, PAM, BIPQ) have been validated. The translated questionnaires require a process of validation. If the Nepalese questionnaires were used and validated, it has to be cited and referenced and all interviews to be conducted by referring to the Nepalese version questionnaires copies.

Statistical analyses

Page 7 Line 221-222, proper citation including publisher name to be provided.

Page 7 Line 225, the actual Step wise method and actual name of the statistical test (e.g. whether via GLM, logistic regression tab etc) in the SPSS to be stated.

Page 7 Line 226, coding for category HL and PA levels to be stated. Sub domains to be stated if analyzed.

Page 7, information on missing data to be reported if any.

Results

Table 1, uneducated and educated to be clearly defined. The age category interval inconsistent and how are they categorized? HPS, HIS, SS, AE, UHI and PAM to be denoted in the table footnote. Title to include the word baseline. Table requires cosmetic improvement.

Table 2, the presentation of the table requires improvement including cosmetic changes with the p value in the last column and presented in landscape form if space is limited. n (%) to be labelled. The two rows of Total illness perception scale to be merged and clearly separated from other rows. Technically p value cannot be zero (to use symbol < ). HPS, HIS, SS, AE, UHI and PAM to be denoted in the table footnote.

More detail information on the estimates, model summary/fit, multicollinearity issue (if any), any adjustment to the p value/alpha level (if any), whether interaction was explored in the analyses to be provided/stated or discussed.

There are numerous typographical errors which require editing.

Here are some examples.

i) Citation to follow journal format i.e. .[ ] etc

ii) US$ to be written as USD

iii) table to be written as Table (standardization purposes)

iv) Line 171, five s

v) Line 200 further. (22, 23).

vi) Line 91, Line 106, Line 222, Line 262, Line 234, Line 294, Line 325 spacing typo e.g. 43%.(5,6). ' (25).Participants Health literacy(26) smoking).The Australia,(35) The Netherlands(36), and USA(37) (43, 44). The

vii) Line 204, products..

viii) ( AOR), space to be omitted.

For those p values with 0.05, another decimal point to be provided to indicate if the p value is equivalent to p=0.05 or smaller/larger than 0.05.

The list of references requires revision and to follow journal format.

7. PLOS authors have the option to publish the peer review history of their article (what does this mean?). If published, this will include your full peer review and any attached files.

Reviewer #3: No

---

## [Author Response · Author response to Decision Letter 1]

20 Feb 2020

Reviewer remarks Authors response 

Page 4 Line 143, the sample size calculation was based on knowledge of self-management. What was the prevalence figure contributed by HL and PA respectively?

 As there was no study on HL and PA in Nepal, and we use the prevalence of self-management in the sample size calculation. This was chosen because both HL and PA contribute to the self-management of conditions.

Page 5, for the subjects selection into the study what was the minimum education attainment and how to ensure that the subjects understand the study and know what they have been asked?. There was no education attainment criterion for subject selection into the study. The study was conducted in a rural setting where the majority of the peoples were uneducated, and the same was revealed from our study (nearly 70%). In order to ensure that the participants can easily understand the study and the employed instruments, a two-day hands-on training was provided to the research assistants [in both Maithili (a local language spoken by chunks in a rural setting) and a Nepali language] in the community setting. This included field activities where they explained the questions and checked understanding. In the pilot, we checked responses for inconsistency and did not find any evidence of inconsistency.

Page 5 Line 145, the total number of RMs in Sunsari District to be stated.

 Six rural municipalities have been added to the updated manuscript 

Page 5 Line 158, what about the level of understanding on the subject matter of the study? The informed consent form was read by the research assistants for those who were unable to read. All of the participants were informed that they were free to withdraw or opt out at any point during the interview. Prior to the interview, written informed consent was obtained from all literate participants, and thumb impressions were obtained from illiterate participants

Page 5 Line 159-160, the sentence can be further improved Study participants with a hearing disability, severe cognitive disorder, kidney disease, a history of stroke or diagnosed heart attack, and with a terminal illness such as cancer were excluded from the study. 

Page 6 Line 210, brief IPQ to be written as BIPQ. In line with your remarks, we have corrected it as BIPQ in the revised manuscript

There was no information if the Nepalese version questionnaires (HLQ, PAM, BIPQ) have been validated. The translated questionnaires require a process of validation. If the Nepalese questionnaires were used and validated, it has to be cited and referenced, and all interviews to be conducted by referring to the Nepalese version questionnaires copies.

 The HLQ and BIPQ have been used in a previous study in Nepal. PAM has been validated, and we are writing a manuscript for PAM validation(level of internal consistency is good (α = 0.88)). 

The link to HLQ and BIPQ study in Nepal:

1. https://www.ncbi.nlm.nih.gov/pubmed/30412262 (HLQ)

2. https://core.ac.uk/download/pdf/145237476.pdf (BIPQ)

All interviews were conducted using Nepalese version questionnaires of HLQ, BIPQ and PAM.

In line with your remarks, we have revised our paper with proper citation.

Statistical analyses

Page 7 Line 221-222, proper citation including publisher name to be provided.

Page 7 Line 225, the actual Step wise method and actual name of the statistical test (e.g. whether via GLM, logistic regression tab etc) in the SPSS to be stated.

Page 7 Line 226, coding for category HL and PA levels to be stated. Sub domains to be stated if analyzed. More detail information on the estimates, model summary/fit, multicollinearity issue (if any), any adjustment to the p value/alpha level (if any), whether interaction was explored in the analyses to be provided/stated or discussed. The statistical analyses were performed using the IBM Statistical Package for Social Sciences (SPSS 23.00)(29). Participant characteristics were summarised as frequency and percentage. Normality of the data was assessed using the Shapiro-Wilk test. The inter-quartile range was calculated for all included HLQ domains separately, where the upper quartile cut-offs were used to define "high health literacy level" and the lower two quartiles as "low health literacy level". The raw scores for PAM were categorized into four activation levels (Level 1, 2, 3, and 4) based on a standard cut-off of PAM where level 1 was labeled as "poor activation stage" and Level 2 to 4 as "activation" stage. We used the Chi-Square test for univariate analysis was conducted for all included five domains of HLQ and the PAM where all the independent variables that had p<0.2 were considered in multivariable regression analysis. We assessed the multicollinearity of covariates using Variance Inflation Factors (VIFs). The VIFs for all covariates that were included in the logistic regression analysis were less than 2.0. Backward elimination step-wise multivariable logistic regression analysis was used to identify the determinants of poor HL and PA levels, and the p-value ≤ 0.05 were considered as significant.

Page 7, information on missing data to be reported if any.

 There was no missing data in this study

Results

Table 1, uneducated and educated to be clearly defined. The age category interval inconsistent and how are they categorized? HPS, HIS, SS, AE, UHI and PAM to be denoted in the table footnote. Title to include the word baseline. Table requires cosmetic improvement.

 Education status is defined in the revised table and age category is revised.

HPS, HIS, SS, AE, UHI and PAM is denoted in the table footnote.

Cosmetic improvement is done.

Table 2, the presentation of the table requires improvement including cosmetic changes with the p value in the last column and presented in landscape form if space is limited. n (%) to be labelled. The two rows of Total illness perception scale to be merged and clearly separated from other rows. Technically p value cannot be zero (to use symbol < ). HPS, HIS, SS, AE, UHI and PAM to be denoted in the table footnote.

 Thank you very much for your sharp look. In line with your remarks, we have revised the manuscript

There are numerous typographical errors which require editing.

Here are some examples.

i) Citation to follow journal format i.e. .[ ] etc

ii) US$ to be written as USD

iii) table to be written as Table (standardization purposes)

iv) Line 171, five s

v) Line 200 further. (22, 23).

vi) Line 91, Line 106, Line 222, Line 262, Line 234, Line 294, Line 325 spacing typo e.g. 43%.(5,6). ' (25).Participants Health literacy(26) smoking).The Australia,(35) The Netherlands(36), and USA(37) (43, 44). The

vii) Line 204, products..

viii) ( AOR), space to be omitted. Typographical errors are corrected and paper is thoroughly edited.

For those p values with 0.05, another decimal point to be provided to indicate if the p-value is equivalent to p=0.05 or smaller/larger than 0.05.

 We have revised the table, and your valuable suggestions are considered in the paper.

The list of references requires revision and to follow journal format Corrected in line with Plos One format

---

## [Decision Letter · Decision Letter 2]

16 Mar 2020

PONE-D-19-18916R2

Levels and determinants of health literacy and patient activation among multi-morbid COPD people in rural Nepal: Findings from a cross-sectional study

PLOS ONE

Dear Mr Yadav,

Thank you for submitting your manuscript to PLOS ONE. After careful consideration, we feel that it has merit but does not fully meet PLOS ONE’s publication criteria as it currently stands. Therefore, we invite you to submit a revised version of the manuscript that addresses the points raised during the review process.

We would appreciate receiving your revised manuscript by Apr 30 2020 11:59PM. To enhance the reproducibility of your results, we recommend that if applicable you deposit your laboratory protocols in protocols.io, where a protocol can be assigned its own identifier (DOI) such that it can be cited independently in the future. For instructions see: http://journals.plos.org/plosone/s/submission-guidelines#loc-laboratory-protocols

We look forward to receiving your revised manuscript.

Kind regards,

Kamal Gholipour, PhD

Academic Editor

PLOS ONE

Reviewers' comments:

Reviewer's Responses to Questions

**Comments to the Author**

1. If the authors have adequately addressed your comments raised in a previous round of review and you feel that this manuscript is now acceptable for publication, you may indicate that here to bypass the “Comments to the Author” section, enter your conflict of interest statement in the “Confidential to Editor” section, and submit your "Accept" recommendation.

Reviewer #3: (No Response)

2. Is the manuscript technically sound, and do the data support the conclusions?

Reviewer #3: (No Response)

3. Has the statistical analysis been performed appropriately and rigorously? 

Reviewer #3: No

4. Have the authors made all data underlying the findings in their manuscript fully available?

Reviewer #3: (No Response)

5. Is the manuscript presented in an intelligible fashion and written in standard English?

Reviewer #3: (No Response)

6. Review Comments to the Author

Reviewer #3: The authors have put in effort to address the comments.

Minor comments

For the logistic regression, information on goodness of fit test/model fit to be stated.

Please explain how the multicollinearity was performed for variables more than 2 categories in SPSS? The coding information for all variables to be provided.

Was interaction explored in the analysis?

7. PLOS authors have the option to publish the peer review history of their article (what does this mean?). If published, this will include your full peer review and any attached files.

Reviewer #3: No

---

## [Author Response · Author response to Decision Letter 2]

25 Mar 2020

Reviewer remarks Authors response 

For the logistic regression, information on goodness of fit test/model fit to be stated. Thank you very much. Have stated in the statistical analysis section:

The Hosmer-Lemeshow test was used for goodness of fit in the logistic regression model.

Please explain how the multicollinearity was performed for variables more than 2 categories in SPSS. The coding information for all variables to be provided 

We choose a reference category with a larger fraction of the cases and have created a dummy variable for a viable that represent a categorical variable with three or more categories.( https://statisticalhorizons.com/multicollinearity) 

Coding Information are:

HPS: 0: high, 1: low

HIS:0: high, 1: low

SS: 0: high, 1: low

AE: 0: high, 1: low

UHI: 0: high, 1: low

PAM: 0: high, 1: poor

Gender: Male:0, Female:1

Age category: ≤40:0, 41-55:1, ≥56:2

Education status: Educated:0, Uneducated:1

 Marital status: Married:0, Others:1

Ethnicity: Higher Caste:0, Indigenous:1, Dalits:2

Occupation: Professional:0, Farmer:1, Factory worker:2

Family level income: <20,000: 1, >20,000:0

Use of tobacco products: No:0, Yes:1

Presence of co-morbidity: At least one: 1, Two or more:0

Was interaction explored in the analysis? Yes, we explored the interaction and could not find any.

---

## [Decision Letter · Decision Letter 3]

7 May 2020

Levels and determinants of health literacy and patient activation among multi-morbid COPD people in rural Nepal: Findings from a cross-sectional study

PONE-D-19-18916R3

Dear Dr. Yadav,

We are pleased to inform you that your manuscript has been judged scientifically suitable for publication and will be formally accepted for publication once it complies with all outstanding technical requirements.

With kind regards,

Wen-Jun Tu

Academic Editor

PLOS ONE

Additional Editor Comments (optional):

Reviewers' comments:

Reviewer's Responses to Questions

**Comments to the Author**

1. If the authors have adequately addressed your comments raised in a previous round of review and you feel that this manuscript is now acceptable for publication, you may indicate that here to bypass the “Comments to the Author” section, enter your conflict of interest statement in the “Confidential to Editor” section, and submit your "Accept" recommendation.

Reviewer #3: (No Response)

2. Is the manuscript technically sound, and do the data support the conclusions?

Reviewer #3: Yes

3. Has the statistical analysis been performed appropriately and rigorously? 

Reviewer #3: (No Response)

4. Have the authors made all data underlying the findings in their manuscript fully available?

Reviewer #3: Yes

5. Is the manuscript presented in an intelligible fashion and written in standard English?

Reviewer #3: Yes

6. Review Comments to the Author

Reviewer #3: The Hosmer-Lemeshow test outcome i.e p > 0.05 (indicating model is fit) to be denoted in the result section.

7. PLOS authors have the option to publish the peer review history of their article (what does this mean?). If published, this will include your full peer review and any attached files.

Reviewer #3: No

---

## [Editor Report · Acceptance letter]

12 May 2020

PONE-D-19-18916R3 

Levels and determinants of health literacy and patient activation among multi-morbid COPD people in rural Nepal: Findings from a cross-sectional study 

Dear Dr. Yadav:

I am pleased to inform you that your manuscript has been deemed suitable for publication in PLOS ONE. Congratulations! Your manuscript is now with our production department. 

With kind regards,

on behalf of

Dr. Wen-Jun Tu 

Academic Editor

PLOS ONE